

# Ordovician to Silurian graptolite specimen images for global correlation and shale gas exploration

Hong-He Xu [1*], Zhi-Bin Niu [1,2*], Yan-Sen Chen [1], Xuan Ma [1], Xiao-Jing Tong [1], Yi-Tong Sun [1], Xiao-Yan Dong [1], Dan-Ni Fan [1], Shuang-Shuang Song [1], Yan-Yan Zhu [1], Ning Yang [1], Qing Xia [1]

[1] State Key Laboratory of Palaeobiology and Stratigraphy, Nanjing Institute of Geology and Palaeontology and Center for Excellence in Life and Paleoenvironment, Chinese

Academy of Sciences, 210008 Nanjing, China

[2] College of Intelligence and Computing, Tianjin University, 300354 Tianjin, China

*The authors contributed equally to this work.

**Correspondence**: Hong-He Xu (hhxu@nigpas.ac.cn), or Zhi-Bin Niu (zniu@tju.edu.cn)

## Abstract

Multi- elemental and -dimensional data are more and more important during the development of data-driven research, as is the case in modern palaeontology, in which visual examination, by experts or someday the artificial intelligence, to every fossil specimen acts a crucial and fundamental role. We here release an integrated image dataset of 113 Ordovician to Silurian graptolite species or subspecies that are significant in global stratigraphy and shale gas exploration. The dataset contains 1550 high-resolution graptolite specimen images and scientific information related to the specimen, e.g., every specimen's taxonomic, geologic, geographic, and related references. We develop a tool, FSIDvis (Fossil Specimen Image Dataset Visualiser), to facilitate the human-interactive exploration of the rich-attribution image dataset. A nonlinear dimension reduction technique, t-SNE (t-Distributed Stochastic Neighbor Embedding), is employed to project the images into the two-dimensional space to visualise and explore the similarities. Our dataset potentially contributes to the analysis of the global biostratigraphic correlations and improves the shale gas exploration efficiency by developing an image-based automated classification model. All images are available from https://doi.org/10.5281/zenodo.5205216 (Xu, 2021).



## 1. Background



Graptolite is marine colonial organic-walled hemichordate and has over
210 genera/3,000 species worldwide fossil records, extending from the
Cambrian to the Carboniferous (c. 510~320 Ma) shale sediments (Maletz,
2017). Graptolite extensively diversified in the Ordovician and witnessed the
second-largest mass extinction in geological life history, i.e., the end-
Ordovician mass extinction (Goldman et al., 2020). Graptolite evolved quickly
and spread globally in the Paleozoic (Fig. 1); therefore, its species are widely
used as significant index fossils for determining rock ages and regional bio-
stratigraphical correlations. Graptolite bio-zones divided the Ordovician and
Silurian sediments are generally less than one million years in duration; such
a short geological moment makes it possible for a better and accurate
understanding of the stratigraphy and ancient life macro-evolution (Chen et
al., 2012; 2018). Up to 102 Ordovician and Silurian graptolite species were
selected as global bio-zones for dating rocks, biostratigraphy, regional
correlation, and understanding the evolutionary patterns of ancient life; and 13
global stratotype section and point (GSSP) have been defined by the first
appearance datum (FAD) of graptolite species in the Lower Paleozoic, i.e.,
Cambrian, Ordovician, and Silurian (Goldman et al., 2020). Two of these
GSSPs are situated in southern China (i.e., the bases of the Darriwilian in the
Middle Ordovician and Hirnantian in the Late Ordovician) (Goldman et al.,
2020; Zhang et al., 2020) (Fig. 2).
Additionally, bio-zones or indication zones based on graptolite species
assist with identifying mining beds for shale gas exploration (Fig. 1). Graptolite
shale comprises more than 9% of hydrocarbons rocks and yields the most
significant volume of shale gas globally (Klemme and Ulmishek, 1991;
Podhalańska, 2013). In China, over 61.4% of the natural gas is yielded from
the Ordovician and Silurian graptolite shale of southern China (Zou et al.,
2019). Identification of graptolite species helps to locate shale gale mining
beds; especially, 16 graptolite species were chosen as "gold caliper" to locate
favourable exploration beds (FEB) of shale gas in China (Zou et al., 2015)
(Fig. 2).
In this paper, we release a unique graptolite image dataset, which
consists of 113 key graptolite species used for dating rocks, global correlation,



and "gold caliper" for locating shale gas FEBs in China. All images were taken
from 1,550 carefully curated graptolite specimens collected from the
Ordovician to Silurian sediments of China. We incorporated revision
suggestions from distinguished palaeontologists to generate the ground-truth
labels, providing a taxonomical authority of the dataset. The dataset
potentially contributes to a range of scientific activities and provides 1) an
easy access to high-resolution images of 1,550 specimens of 113 graptolite
species for teaching and training in palaeontology and geologic survey; 2)
global bio-stratigraphical correlation using graptolites, especially with those
bio-zone species; 3) a standard fossil specimen image dataset used in shale
gas industry to improve exploration efficiency, and 4) the potential aid of
developing image-based automated classification model.

**2. Materials and methods**
Images of our dataset were taken from 1,550 graptolite specimens, which
taxonomically belong to 113 graptolite species or subspecies. These
specimens are preserved as shale and were collected from 154
representative geological sections of China. All specimens are housed at the
Nanjing Institute of Geology and Palaeontology (NIGP), Chinese Academy of
Sciences (CAS), the world's largest palaeontological research centre, and one
of the top three specimen collection centres. The NIGP-CAS hosts over 180
palaeontological researchers and laboratory technicians and collecting over
800,000 pieces of fossil specimens from all around the world since 1928
(NIGP, 2011).
Every piece of the specimen is tagged with information, including scientific
names (genus and species names), nominator, nomination year, specimens'
serial number, collection-number, locality (province, city, county), geological
horizon and section, collector name, collecting time, identifier, identifying time,
related references, published illustrations. Specimens can be indexed and
located in their detailed housing drawers and cabinets using any of the above
information. Their detailed research-related information can also be obtained
from the geological section-based database, the Geobiodiverisy Database (Xu
et al., 2020). All this related information is collected and recorded in a
separate spreadsheet file released with our image dataset.
We spent over two years to complete photographing every specimen



using a single-lens reflex camera Nikon D800E with Nikkor 60 mm macro-lens
and Leica M125 and M205C microscopes equipped with Leica cameras (Fig.
3). Every image is well focused and better shows the morphology of graptolite
bodies. In total, we took 40,597 images, including 20,644 camera photos
(each with a resolution of 4,912 × 7,360) and 19,953 microscope photos (each
with a resolution of 2,720 × 2,048). Photos of low contrast or bad focus were
removed from the whole collection. We only kept and selected the photos that
show the visual morphology of every specimen and the diagnostic character
of each graptolite species that the specimens represent (Fig. 4). We selected
one image for each specimen as the present final dataset, uploaded to, and
stored in our cloud server (Fig. 3).
Considering some of the specimens of our collection have a long research
history since 1958, and their taxonomical status might change in the new light
of graptolite systematic study (Maletz, 2017; Zhang et al., 2020), we invited
graptolite palaeontologists to curate every specimen to make sure that its
scientific information is updated and widely accepted.

**3. Data description**
Our dataset consists of 1,550 high-resolution images and a related
spreadsheet file. Every image is a high-resolution photo taken from the
collection of 1550 graptolite specimens. These specimens were formally
published in 1958-2020, and taxonomically belonging to 113 graptolite
species or subspecies, of 41 genera and 16 families of the Order
Graptoloidea (see the uploaded spreadsheet file, Fig 5). The geological age of
these graptolite species ranges from the Middle Ordovician to (467.3 Ma) to
the Telychian (433.4 Ma) of the Silurian period (Fig. 5).
These graptolite species have relatively abundant fossil records and are
significant in regional and global bio-stratigraphical correlations and locating
favourable exploration bed (FEB) of shale gas in China. They are commonly
used in geological age determination and shale gas FEB indication, including
32 graptolite biozones from the Darriwilian stage of the Ordovician (467.3 Ma)
to the Telychian stage of the Silurian (433.4 Ma) and 16 "gold callipers" of
shale gas favourable exploration beds (FEBs) for cases of 20 to 80 m thick
graptolite shale in China (Fig. 6). These species also include two "golden
spike" graptolite species for the two GSSPs in southern China (i.e., bases of





the Darriwilian in the Middle Ordovician and Hirnantian in the Late
Ordovician).
The name of the individual image file is initialled by the specimens' unique
number and then its taxonomical species name. The image file is in JPG
format, and the single JPG file size ranges from 840 KB to 10.59 MB. The
whole volume of the dataset is 6.41 GB.
In the spreadsheet file, we incorporated revision suggestions of several
distinguished palaeontologists for the authority of the graptolite taxonomy. The
spreadsheet file shows the detailed scientific information of every graptolite
specimen. The spreadsheet file includes following fields: species ID, Phylum,
Class, Order, Suborder, Infraorder, Family, Subfamily, Genus, Revised
species name, tagged species name, total number of specimens, specimens
serial number, image file name, microscope photo numbers, SLR photo
number, Stage, Age from, Age to, mean age value, Locality, Longitude,
Latitude, Horizon, and specimens firstly published reference.

**4. Data visualization**
We have developed an interactive web exploration tool, FSIDvis (Fossil
Specimen Image Dataset Visualiser), to assist users to examine better the
scientific contents of our data (Fig. 7).
We further explore the distribution of these graptolite images and
visualize the t-SNE feature embedding of our graptolite dataset (Fig. 8) using
different colors to denote different families. In detail, for each annotated
image, we first resized it into 448×448 pixels and fed it into the trained CNN
model. The output 1×1×2048 feature map from the last average pooling layer
is flattened and projected to a 113 (number of species) dimensional fully
connected layer to represent an image embedding. After that, we use t-SNE
(t-Distributed Stochastic Neighbor Embedding), a nonlinear dimension
reduction technique for high-dimensional data, to project the image
embeddings into the two-dimensional space for visualization. Finally, we
indicate the image data distribution by a scatter plot, we use 15 colors to
represent 15 families of the order Graptoloidea, covering 42 genera and 113
species, so the distribution of the images in this figure is based on species,
which shows a "big mixed, small settlements" posture.



### 5. Conclusions

A graptolite specimen image dataset containing 1550 high-resolution images is released. The formation of our dataset includes two steps. 1) 113 Ordovician to Silurian graptolite species or subspecies are selected for their significances in global stratigraphy and shale gas exploration; 2) 1550 pieces of fossil specimens that typically represent these 113 species are carefully curated and photographed.

Scientific information related to these graptolite specimens is also included and recorded for further study. The structured records include taxonomical, geologic, geographic, and related references of every specimen.

Our dataset potentially contributes to global bio-stratigraphical correlation, especially with those graptolite bio-zone species, in the shale gas industry to improve exploration efficiency and develop an image-based automated classification model.

The whole dataset has visualised the tool FSIDvis (Fossil Specimen Image Data Visualizer). A nonlinear dimension reduction technique, t-SNE (t-Distributed Stochastic Neighbor Embedding), is used to our data and project the image embeddings into the two-dimensional space for visualisation.

**Data availability.** The dataset is archived and publicly available from https://doi.org/10.5281/zenodo.5205216. Visualized version is available at https://fossil-ontology.com/FSIDvis/graptolite/.

**Author contributions.** H.-H.X. and Z.-B.N. equally designed the project, developed the model, and performed the simulations. H.-H.X. prepared the manuscript with contributions from Z.-B.N. Y.-S.C. gave technician supports. X.M. revised and examined fossil specimens. Others contributed in specimen photography.

**Competing interests.** The authors declare that they have no conflict of interest.

**Acknowledgments.** We thank Dr. Pan Zhaohui, Institute of Vertebrate Paleontology and Paleoanthropology, CAS; Mr. Pan Yaohua and Mr. Wu Junqi, College of Intelligence and Computing, Tianjin University, for



constructive suggestions and help.

**Financial support.** This research has been supported by the Strategic
Priority Research Program of the Chinese Academy of Sciences (Grants
XDA19050101 and XDB26000000) and National Natural Science Foundation
of China (Grants 41772012 and 61802278).

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

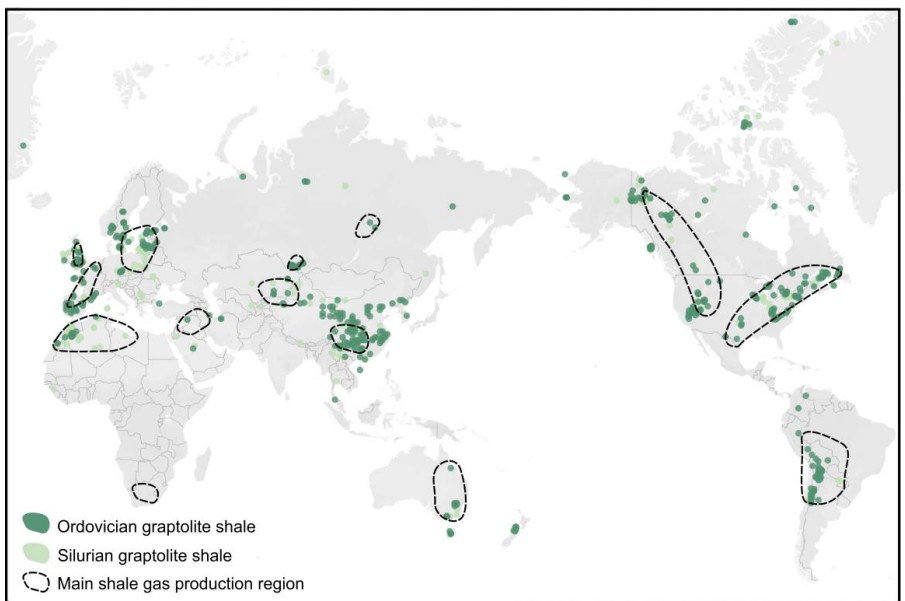


**Figure 1.** Global distribution of graptolite shales and shale gas production

region. Most graptolites were yielded from the shale and their distribution is

based on graptolite fossil occurrence records in global Ordovician and Silurian

sediments. All data are from Peters and McClennen (2016) and Xu et al.

(2020). Graptolite shale comprises over 9% of hydrocarbons rocks in the

world and yields the largest volume of shale gas in the world. In China, over

61.4% natural gas was yielded from the Ordovician and Silurian graptolite

shales of southern China. The map is from © OpenStreetMap contributors

2021. Distributed under the Open Data Commons Open Database License

(ODbL) v1.0.

272



273

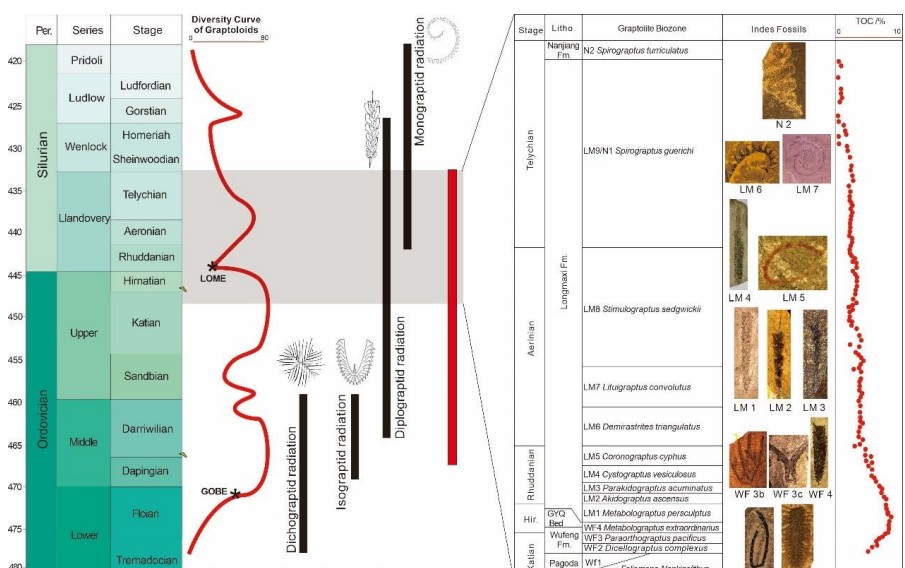

274

**Figure 2.** Geological significance and application of graptolites. Our dataset of
graptolites is significant to biostratigraphy and the dating of the Ordovician
and Silurian sediments. They are widely distributed around the world and
useful for regional correlation. These graptolites have also witnessed several
macro-evolutional events, including the great Ordovician biodiversity event,
Late Ordovician mass extinction, radiation in several graptolite groups, and
global stratotype section and point (GSSP), based on graptolite species. To
date, 13 GSSPs have been defined by the FAD of graptolites in the early
Paleozoic. Two are in South China (i.e., the bases of the Darriwilian in the
Middle Ordovician and Hirnantian in the Late Ordovician) (the spike marks in
the figure) (data from Goldman et al., 2020). Bio- or indication zones based on
graptolite species assist with identifying mining beds for shale gas exploration
in southern China. 16 graptolite indicator-zones are used in the shale gas
exploration in China (Zou et al., 2015).



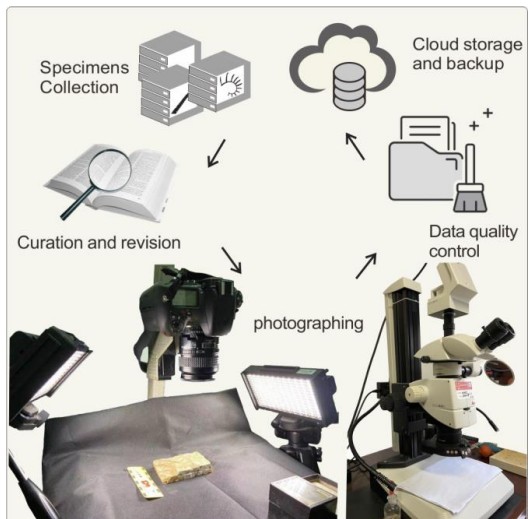

290

**Figure 3.** The process of creating the graptolite specimen image dataset.
The graptolite specimens were carefully curated and revised to select the
species with biostratigraphy and application significance. Every image was
obtained from specimens that were macro-photographed using a single-lens
reflex camera and microscope. After professional revision and cleaning, the
whole dataset was uploaded to and stored in our cloud server.

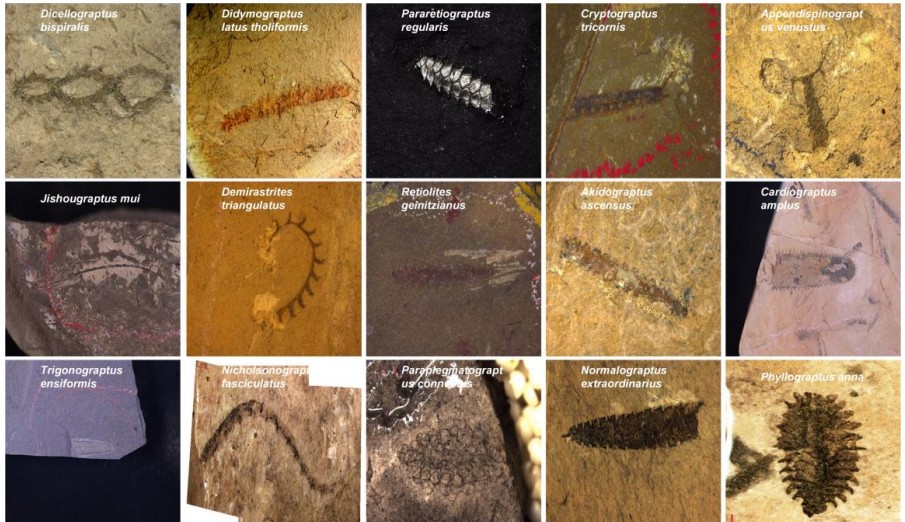

298

**Figure 4.** Typical images of our dataset. Every image was taken from a
unique graptolite specimen. Photos of low contrast or bad focus were
removed. Our dataset only selected the photos that well show visual
morphology of every specimen and diagnostic character of each graptolite
species that the specimens represent. The scientific species name of every
specimen is given on each image.



306

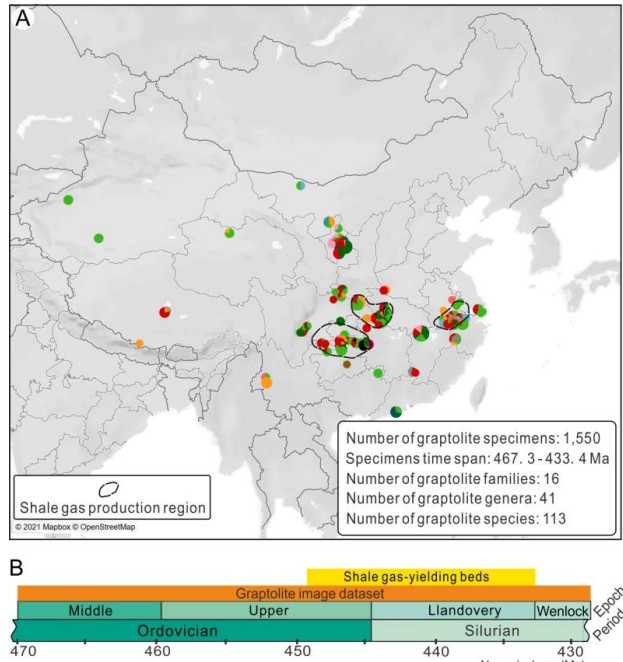

307

**Figure 5.** Geographic distribution (A) and geologic range (B) of graptolite species of our dataset. Each graptolite specimen locality is represented by a pie chart where each colour is encoded as one graptolite family of the order Graptoloidea. The sector size is proportional to the specimen number for every family. The radius of the pie chart is proportional to the total number of specimens from the same locality. The dashed-lines circle the main areas of shale gas production. The map is from © OpenStreetMap contributors 2021. Distributed under the Open Data Commons Open Database License (ODbL) v1.0.









| System | Series | Stage | Graptolite biozone (22) |
|---|---|---|---|
| Silurian | Wenlock | Homerian | *Colonograptus deubeli* |
| | | | *Colonograptus praedeubeli* |
| | | Sheinwoodian | |
| | Llandovery | Telychian | *Spirograptus turriculatus* |
| | | Aeronian | *Lituigraptus convolutus* |
| | | | *Demirastrites triangulatus* |
| | | Rhuddanian | *Coronograptus cyphus* |
| | | | *Cystograptus vesiculosus* |
| | | | *Parakidograptus acuminatus* |
| | | | *Akidograptus ascensus* |
| Ordovician | Upper | Hirnantian | *Metabolograptus persculptus* |
| | | | *Metabolograptus extraordinarius* |
| | | Katian | *Paraorthograptus pacificus* |
| | | | *Dicellograptus complexus* |
| | | | *Dicellograptus ornatus* |
| | | | *Dicellograptus complanatus* |
| | | Sandbian | *Orthograptus calcaratus* |
| | Middle | Darriwilian | *Hustedograptus teretiusculus* |
| | | | *Archiclimacograptus riddellensis* |
| | | | *Pterograptus elegans* |
| | | | *Nicholsonograptus fasciculatus* |
| | | | *Levisograptus dentatus* |
| | | Dapingian | *Levisograptus austrodentatus* |

| Stage | Graptolite indicator zone for shale gas FEB (16) |
|---|---|
| Telychian | *Spirograptus turriculatus* (N2) |
| | *Spirograptus guerichi* (N1) |
| Aeronian | *Stimulograptus sedgwickii* (LM8) |
| | *Lituigraptus convolutus* (LM7) |
| | *Demirastrites triangulatus* (LM6) |
| Rhuddanian | *Coronograptus cyphus* (LM5) |
| | *Cystograptus vesiculosus* (LM4) |
| | *Parakidograptus acuminatus* (LM3) |
| | *Akidograptus ascensus* (LM2) |
| | *Metabolograptus persculptus* (LM1) |
| Hirnatian | *Metabolograptus extraordinarius* (WF4) |
| Katian | *Dicellograptus mirus* (WF3c) |
| | *Tangyagraptus typicus* (WF3b) |
| | *Paraorthograptus pacificus* (WF3a) |
| | *Dicellograptus complexus* (WF2) |
| | *Dicellograptus complanatus* (WF1) |


**Figure 6.** Graptolite species selected as global biozone (left) and indicator
zone (right) for shale gas favourable exploration beds of our dataset. Among
our dataset of 113 graptolite species, there are 22 graptolite index species
from global correlation from the Middle Ordovician to (470.0 Ma) to the
Wenlock of Silurian period (427.4 Ma), and 16 graptolite species as 'gold
calliper' to locate favourable exploration beds (FEB) of shale gas in China.
Note that some graptolite species are duplicate in the two lists.


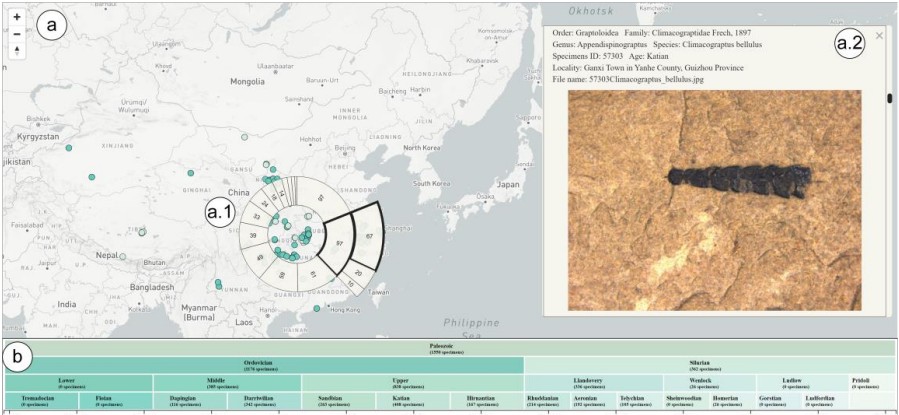


**Figure 7.** FSIDvis (Fossil Specimen Image Dataset Visualiser) system
interface. a) Fossil map view. a.1, is a tailor-designed specimens' picker that
facilitates users to collect interest fossils of a region where the inner ring and
outer ring represent the family and genus. When the user chooses a genus,
the corresponding detailed species with images will be listed in the a.2 view.
b) Time view, providing the time selection ability; the top one is the
chronostratigraphic time scale, and the bottom one is a time slider that
facilitates the users to choose a specific time slot interactively. The web
exploration tool of graptolite is provided at https://fossil-
ontology.com/FSIDvis/graptolite/. The map is from © OpenStreetMap
contributors 2021. Distributed under the Open Data Commons Open
Database License (ODbL) v1.0.

342



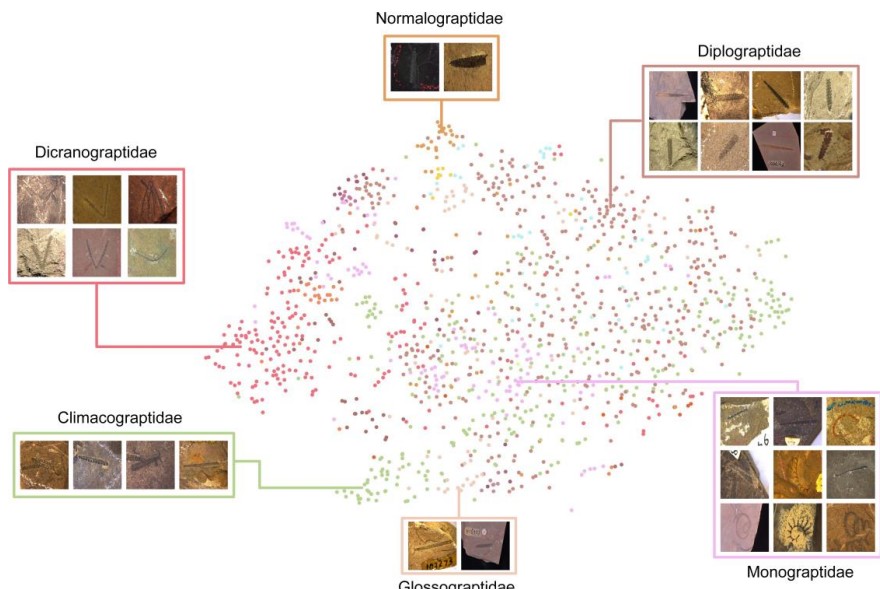

343

**Figure 8. t-SNE embedding visualization of our image dataset.**