# Peer review of "Ordovician to Silurian graptolite specimen images for global correlation and shale gas exploration"

_Earth System Science Data, 2021_

## Referee Comment (RC1)

Manuscript:   essd-2021-280
Authors:      Xu, H.-H., Niu, Z.-B., Chen,Y.-S.,  et al.
Title:        Ordovician to Silurian graptolite specimen images for global correlation and shale gas exploration.

**Summary:**       It is a welcome development that more museums are placing photographs of their fossil collections onto the internet as publicly accessible galleries.  This allows researchers to determine whether to travel to view the physical specimens – often an expensive undertaking – or to seek to loan the specimens, which can risk damage or loss of irreplaceable materials.  This contribution will make available many high-resolution images of graptolite specimens from 154 important Chinese locations below and above the Ordovician Silurian boundary.  Graptolites are one of three index fossil groups (graptolites, conodonts and chitinozoans) most widely used for global correlation in this time interval. The same interval includes a major extinction event across which the proportions of trilobite families change dramatically.  The host shales are an important source rock for gas.  The gallery of images has value for the local economic geologists and for academic research.

The fossil collection is connected to publications spanning many years.  Fortunately, the authors have sought expert help to ensure that the graptolite taxonomy is up to date.  Unfortunately, the experts are not all named.  The images are accessible through five large ZIP files.  A single Excel file provides detailed records for the 1550 high-resolution images of 113 species and subspecies.  This file provides both the current expert name for the taxon and the originally "tagged" name in the collection.  It includes higher taxonomic categories, collection locality, specimen number, image file name and publication.  It also includes ages for the ends of a range.  It does not make clear whether this is an estimate of the global range, the local range, or the age of the photographed specimen.  The "mean age" column is rather puzzling, unless the values are attempts to calibrate the age uncertainty for the actual specimen (this seems unlikely).  Of course, these calibrations are subject to change as new dates emerge, for example from the 2012 to the 2020 editions of Elsevier's Global Time Scale books.

Downloading the ZIP and xslx files is straightforward.  Finding the ZIP file for a given taxon image is very cumbersome.  Surely the ZIP file could have a column to show this.

The abstract claims to support an image-based automated classification model.  I am unable to find justification for this claim, perhaps because it was not possible to access the necessary tools with the url provided.  It was not clear to me how readers would determine the original dimensions of the specimens in the images that we have downloaded.  No image that I examined has included a scale. This is a very basic requirement for illustrations of paleontological specimens.  Line 165 mentions resizing the images.  Perhaps all would be clear if it were possible to access the FSIDvis tool.

I was unable to access the FSIDvis tool from the information given.  The text figures explained a lot, but lines 158-175 include some technical details about t-SNE that I could not understand.

The text includes numerous common failures of translation from Chinese to English.  They are distracting, but almost none obscure the meaning.

**Data and Tool Accessibility:**      I downloaded one of file large ZIP files that make up the gallery (272images_4.zip).  It took one hour, using a fast internet connection at my university to download the 1.3GB.  I examined a selection of the images.  The photography is of high quality.  The fossil preservation

is not the best, but that is presumably a true representation of the nature of the material at these localities. Some of the images included specimen numbers. None of the images included a scale. The images are valuable for anyone doing local or inter-regional research that includes the Chinese Lower Paleozoic. Expert users can make their own judgment about the species assignments, but few of the specimens could be recommended to those who wish to learn the characteristics of a taxon. For that, the original holotypes are essential and better-preserved material is desirable.

The xlsx file seems not to indicate which of the five ZIP files contains a given image. Instead, the "Preview" button on the host xenodo website opens a list of the image file names, for a given ZIP file. Most file names begin with the specimen number, followed by the species binomen. Some specimen numbers are alphanumeric. It is not easy to predict the range of specimen/image numbers from one ZIP file to the next. Surely the authors could add to their useful xlsx file a column indicating which ZIP file holds the image.

The caption to text-figure 7 refers to a "Fossil Specimen Image Dataset Visualiser" (FSIDvis) with a clickable link. The link is compromised by a line break. I tried entering many versions of the link to remove the page break. All yielded either an error message or an unsuccessful Google search. Another link appears at the end of the text under "Data Availability." It did not generate an error, but leads to a blank pale-blue screen. Consequently, it is not possible to comment on the quality or utility of this tool. Access to it might have overcome some short comings of the text and images.

**Text Style:** The usual Chinese-to-English grammatical errors are present. Two are most common: mismatches between singular are plural for verbs and subjects; and the misuse/omission of the definite and indefinite articles (the / a / an / some). The opening line (39) has examples of both flaws. I understand that Mandarin has no grammatical equivalents. Any English-speaking editor could easily fix all this. A spell-checker would find 'Geobiodiverisy' (line 103). More difficult would be the apparent contraction of 'shale gas' to 'gale.'

**Text Content and Omissions:** There may be very few readers who will need the introductory generalizations about graptolite paleontology. Readers from some institutions and museums may be upset by the gratuitous boast about the largest paleontology research center in the world.

Figure 4 claims that the "scientific species name of every specimen is given on each image." This is true for the fifteen images in figure 4, but it is not true for any of the images that I downloaded and viewed. The taxon names appear in the file names, NOT on the images themselves.

What is meant by a "big mixed, small settlements" posture (line175)?

Some readers might like to know the names of the "several distinguished palaeontologists" (lines 148-9) who provided the current taxonomic assignments for the specimens. Did they simply provide current names for older synonyms or did they make some taxonomic revisions for specimens that were initially mis-identified? The experts who performed this service were not named in the xlsx file. The acknowledgements paragraph mentions only one (X.M.). Should readers assume others are among the coauthors? The statement that authors contributed equally to this work (line 13) is not helpful and seems at odds with lines 200-204. Perhaps we should think about it this way: if a figured specimen were to be included in a synonymy list, should all these co-authors be named? Should this contribution to be understood to include any formal re-identifications?

---

## Referee Comment (RC2)

Overall evaluation.

Overall, this is an excellent step forward in terms of providing these resources to the broader scientific community in an interactive and accessible format. I look forward to it being published with the recommended revisions.

I have read the manuscript as well as the posted comments, including those from reviewer #1. I agree with almost all of the comments made by that reviewer, so I will not repeat all of them here. I am pleased to see that the authors have already begun the process of making the necessary revisions.

As noted by the other reviewer, if this paper is to be published it will require very extensive editing to correct the grammar. As a result, I have not made any specific comments related to spelling, grammar, etc.

Specific comments.

I wish to reinforce the point that scales on all of the images are essential and I am glad that this is being changed. As of February 3, I still did not see scales on the images that I looked at either in the database folders or accessed through the FSIDvis tool.

I also had difficulty accessing the FSIDvis tool at fossil-onotology.com, but eventually I got there. However, I could not figure out how to access the images linked to the localities, so I was happy to see that this was explained in the reply to the other reviewer – by hitting the spacebar. This should be explained clearly within the FSIDvis tool web page and it should also be explained in this manuscript.

There are several places in the manuscript where the information about the number of species and images is repeated. Some of this repetition should be removed to make the paper more concise.

Once I was able to access the image files in the FSIDvis tool I was surprised at the naming conventions for the species. The image files that are called up are unusual because, for example, one species that I found was named Climacograptus angustatus but beside that it said that the genus was Proclimacograptus. I presume this is the result of taxonomic revision, but it is confusing, especially to someone less familiar with graptolites. What happens when the species name had also been revised from the original identification, is that also shown in these pop-up image files within the FSIDvis tool? I think it would be better if these pop-up files showed both the original name and the full revised name of each taxon. It would also be helpful if the reference source for the revised name was provided, either in these pop-up files or in the excel file, or both.

Once you have selected a pop-up image within the FSIDvis tool you can click on the image and another copy of the image alone will appear beside the pop-up box. This is good. It would be

even better, however, if, once the full image appeared, if the user could zoom in on it. Many of the photos are made up mostly of surrounding rock and empty space very small images of the actual graptolite specimens, so it is necessary to zoom in to see the actual morphology of the specimen (more on this below). Unfortunately, I was not able to do this within the FSIDvis tool. I could do it if I went to the full database of images and opened each one but, as noted by the other reviewer, it is very to find particular images within that database because the Excel file does not say what folder they are in.

As noted above, many of my concerns about this manuscript were already pointed out by reviewer #1 and I am pleased to hear that the authors are addressing these. However, I do have one additional, very important point and that is related to the quality of many of the images themselves. Many of these specimens should have been photographed at much higher magnification. For example, the first image in folder 1 is specimen 83260 and the specimen occupies only about 10% of the total field of view. There is no reason to have so much wasted space in images such as this, which just means that the image itself is shown at lower resolution than necessary. I think in every possible case the graptolites should have been photographed in the microscope rather than with a camera with a macro lens and the graptolite should fill the field of view as much as possible. The next image, 83269 is much better, although it is hard to see the thecal details as a result of the preservation. The next image, 90359, is not good at all. I cannot even tell where the specimen is that is supposed to be depicted in this image. In fact, there are a number of cases in which the specimens are quite hard to see in the images. Image 21217 illustrates another problem with some of these images. The actual specimen is so small compared to the image frame that by the time I zoom in far enough to see the thecae, the image is too blurry or pixilated to clearly see the thecal form. The same problem exists with image 53891 – by the time I zoom in to see the critical details of the proximal end it is too blurry to see them. Thus, this and many of the other images do not show the critical morphological details needed to identify the species, as suggested in the text of the manuscript. Another example of a different problem with some of the images is specimen 10335 in folder 1. In this case all that can be seen is some generally archiclimacograptid distal thecae. This is definitely not enough information to identify this or any other specimen, on its own to the level of species or maybe even genus, because this could also be a distal specimen of Pseudoclimacograptus. In the case of image 10336, the whole image is blurry and it is not at all clear which of the several specimens in the picture is the one this image is intending to show. Overall, then, higher magnification photographs, with scales, and more careful manipulation of the lighting to enhance the contrast between the specimens and host rock could improve many of these images considerably. This is a good database of images to have available and many of the images are excellent, but its value is considerably weakened by the low magnification and quality of a significant proportion of the photos. I hope that the authors are able to rectify these problems although I expect it will be a very time-consuming effort.

Note that it is because of the relatively poor quality of a significant proportion of the images that I rated the data quality as only good. If I had the choice I would rate it as somewhere between fair and good.

On a side point, I am surprised that the authors say that a scale in the images was not necessary for the AI species recognition. Without knowing the scale how could an AI distinguish two species that have the same thecal and rhabdsosomal form and differ only in width and thecal spacing?

---

## Author Response (AR1)

Revision notes

Manuscript title: Ordovician to Silurian graptolite specimen images for global correlation and shale gas exploration

Corresponding author: Hong-He Xu (hhxu@nigpas.ac.cn)

Comments from two reviewers are all considered and reflected in the revised manuscript. The main problem comprises two aspects, the dataset quality and the tool. We made the revision accordingly.

(1).The dataset quality, improvment

The dataset is updated and the newly uploaded to share. A new DOI is given in the revision. https://doi.org/10.5281/zenodo.619494

We re-organized all images and assured that every specimen image shows with the scale in its own photo or separated photo. Now the updated dataset includes 2951 images and the whole size is 10.4 G. every scaled photo file is named with a postfix S. The image name consists of the number of specimen and the species name. one only trouble is that the volume of the whole dataset is quite large. It takes a while to upload and download all files.

Now it is ensured that every specimen is shown with additional scale bar and that any measurement to the fossil is available.

The reviewer also stated that some images are not quite clear, or some images seem too empty. These are all problem of image quality. The new dataset, with more hi-resolution images and scale, some microscope images were selected for the case that the fossil itself is small but the whole specimens is quite large.

(2).The visualization tool

The FSIDvis is an interactive visual explorer of graptolite specimen image data (FSIDvis), which is accessed through:

http://fsidvis.fossil-ontology.com:8089/

Hit the spacebar to view the details of graptolite specimen. The tool, FSIDvis, is also updated and more detailed instruction of this software is given. the revision, in the fig caption part of this software.

The naming system or the method is explained in the text. Every image file is named after its unique number and labeled species name. When the specimens wad re-studied or transferred to another taxon, the file name does not reflect its taxonomic status. a brief revision record is given in the updated tool, FSIDvis. And I also suggest that readers or users of our dataset check our spreadsheet file for detailed information.

Furthermore, the whole text is revised in text and writing by colleagues. Hoping this version is much better.

Non-related part of fossil specimens collection is deleted as response to the first reviewer. TSNE, or t-SNE (t-Distributed Stochastic Neighbor Embedding), a technique for dimensionality reduction, is particularly well suited for the visualization of high-dimensional datasets that our specimens image dataset belongs to. We tried to analyze these data using this method and show a feasible way to classifying these specimens based on images only.

---

## Referee Report (RR1)

Manuscript Title:
Ordovician to Silurian graptolite specimen images for global correlation and shale gas exploration

Corresponding Author:
Hong-He Xu

REASON TO PUBLISH

As in my first review, I welcome this approach to sharing the contents of paleontological collections in museums and research institutes.  It could save considerable travel costs for research based in part upon remote collections.  It may allow the verification or updating of identifications published in the papers cited in the XLSX file.  I imagine that the quality and quantity of the new images far exceeds that in some of the original publications.  Naturally, they cannot exceed the quality of the specimens themselves.

The accompanying manuscript does not need to be long or detailed.  It is sufficient in its current form.  Most of the data users will likely have their own expert reasons to access the Zenodo database.  The manuscript serves to advertise its existence.  The 5-6 short paragraphs on the Zenodo website are an adequate introduction and well written.  The ESSD manuscript has the added benefit of some references and several helpful illustrations.

AN IMPROVEMENT and A SERIOUS NEW PROBLEM

The revision notes indicate that the authors have now added the essential scale bars that were promised but missing on images in Version 1 (August 16 2021) of the Xenodo data repository.  The new version is presumably enlarged by adding a second set of images that include scales.  Although this would be a very welcome correction, I am still struggling to verify it by opening the Version 2 (Jan 31, 2022) ZIP files.  The downloaded ZIP files are not readable or extractable by my Windows computer.  Neither the file explorer nor the customary unzipping utility can read them.  I have experimented with different utilities and tried renaming the files, but to no avail.  I can still read and extract the Version 1 files.  So, my computer is presumably not the problem.  Until this failing is addressed, there are effectively no scaled images in the database.  It should not be published in this condition.  I would have expected the ESSD staff to at least check that the data are accessible before sending the manuscript for review.  Perhaps they did and this problem simply does not arise for every user.

CONTINUING CONCERNS

The data-rich XLSX file is the index to the image collection.  Surely it should indicate for each taxon or specimen, which of the ZIPPED files contains the corresponding images.  It should not be necessary to download, extract and examine all the ZIP files to find images for a single taxon, publication, or locality.  Perhaps I have somehow overlooked this information.  Downloading the large ZIP files may be slow, but not unreasonable, unless one needs to download all eleven files when perhaps only one is really needed.

There is potential confusion about the Zenodo web address.  In the authors' response it is misprinted – the last digit is missing.  The revised text provides the address in two places, but they are different.  The abstract has been edited to give the new web address for Version 2 of the data.  The "Data Availability" section still has the old address; that is for Version 1 of the database which lacks the scales.

Although the data are surely intended for use by experts, the manuscript includes some very elementary facts about graptolites and Paleozoic stratigraphy. Not all are strictly correct. The local first appearances of graptolite taxa, for example, are a means to correlate with locations away from GSSPs for many Ordovician and Silurian stage boundaries. The graptolite taxa are indicative, not definitive. First appearances may be diachronous and earlier occurrences may even be found at stratotype sections. The "spikes" are definitive, even if less practical.

The various sections of the paper and its figure captions tend to be repetitive and still contain many common errors of English. The repetitiveness is irritating and serves no purpose. The grammatical flaws do not obscure meaning and are not surprising for non-native English-speaking authors. The usual mismatches of singular subjects with plural verbs occur in the first two words of section 1 – "Graptolite was . . " The database managers want readers to trust taxonomic revisions made by un-named experts of their choosing. It would be unfortunate to undermine this by failing to find an English language proof-reader. At one time, trained journal staff would undertake such corrections, but that is less common in large publishing houses today. Fortunately, universities are more cosmopolitan.

REASON TO REQUIRE FUNDAMENTAL REVISION

Given the concerns about quality-control, which range from trivial to nullifying, it is possible that readers will lose trust in the authors' attention to other aspects of their data. Certainly, it begins to shake my confidence in the project. Am I making an elementary unzipping mistake or are the database managers careless? Because the credibility of the data is at risk, it is important to address even minor concerns. I recommend easy remedies for the authors. They should enlist a set of volunteers with different computer expertise and hardware to test the accessibility of their data files. It should also be a simple matter to find a native English-speaking paleontologist who can quickly correct the English grammar and syntax, given an editable text file. I can correct English grammar, but I am worried about database managers and editors whose revised datafiles are unreadable on a very standard Windows computer that can still read the previous version.

Publishing the data in its current format would be a serious disservice to the admirable effort of compiling this large set of illustrations.

---

## Author Response (AR2)

Revision notes

ESSD-2021-280

Manuscript title: Ordovician to Silurian graptolite specimen images for global correlation and shale gas exploration

Corresponding author: Hong-He Xu (hhxu@nigpas.ac.cn)

Revised title: A multi-dimensional dataset of Ordovician to Silurian graptolite specimens images for virtual examination, global correlation and shale gas exploration

Revisions were made based on reviewer's comment and our discussions, which greatly update my understanding of the dataset. They include following aspects.

**REASON TO PUBLISH**

As in my first review, I welcome this approach to sharing the contents of paleontological collections in museums and research institutes. It could save considerable travel costs for research based in part upon remote collections. It may allow the verification or updating of identifications published in the papers cited in the XLSX file. I imagine that the quality and quantity of the new images far exceeds that in some of the original publications. Naturally, they cannot exceed the quality of the specimens themselves.

Author reply:

the potential value of our dataset is shown, the idea of virtual examination to specimens (VES) is proposed and emphasized to the revised manuscript. This is the major revision of the manuscript – a new understanding of the dataset. It is not only a collection of images, but a multi-dimensional data with scientific value.

**AN IMPROVEMENT and A SERIOUS NEW PROBLEM**

The revision notes indicate that the authors have now added the essential scale bars that were promised but missing on images in Version 1 (August 16 2021) of the Xenodo data repository. The new version is presumably enlarged by adding a second set of images that include scales. Although this would be a very welcome correction, I am still struggling to verify it by opening the Version 2 (Jan 31, 2022) ZIP files. The downloaded ZIP files are not readable or extractable by my Windows computer. Neither the file explorer nor the customary unzipping utility can read them. I have experimented with different utilities and tried renaming the files, but to no avail. I can still read and extract the Version 1 files. So, my computer is presumably not the problem. Until this failing is addressed, there are effectively no scaled images in the database. It should not be published in this condition. I would have

Author reply:

The dataset is updated and the newly uploaded to share. According to the editor, we still use the same DOI, for it includes updating. We re-organized all images and assured that every specimen image shows with the scale in its own photo or separated photo. Now the updated dataset includes 2951 images and the whole size is 10.4 G. every scaled photo file is named with a postfix S. The image name consists of the number of specimen and the species name. One only trouble is that the volume of the whole dataset is quite large. It takes a while to upload and download all files.

Now it is ensured that every specimen is shown with additional scale bar and that any measurement to the fossil is available. The visualization tool will be improving for better fulfil the VES.

Although the data are surely intended for use by experts, the manuscript includes some very elementary facts about graptolites and Paleozoic stratigraphy. Not all are strictly correct. The local first appearances of graptolite taxa, for example, are a means to correlate with locations away from GSSPs for many Ordovician and Silurian stage boundaries. The graptolite taxa are indicative, not definitive. First appearances may be diachronous and earlier occurrences may even be found at stratotype sections. The "spikes" are definitive, even if less practical.

Author reply:

This is a good question. Bio-stratigraphy, or correlation using fossil data, so far is still a common and normal method in stratigraphy and GSSP, the work of which is still ongoing globally and approved by international commission on stratigraphy. Of course fossil record might be diachronous in different sediment settings, but such discussion is beyond the scope of this study. We here just show the dataset-concerned fossils have significance in current study.

The various sections of the paper and its figure captions tend to be repetitive and still contain many common errors of English. The repetitiveness is irritating and serves no purpose. The grammatical flaws do not obscure meaning and are not surprising for non-native English-speaking authors. The usual mismatches of singular subjects with plural verbs occur in the first two words of section 1 – "Graptolite was . . " The database managers want readers to trust taxonomic revisions made by un-named experts of their choosing. It would be unfortunate to undermine this by failing to find an English language proof-reader. At one time, trained journal staff would undertake such corrections, but that is less common in large publishing houses today. Fortunately, universities are more cosmopolitan.

REASON TO REQUIRE FUNDAMENTAL REVISION

Given the concerns about quality-control, which range from trivial to nullifying, it is possible that readers will lose trust in the authors' attention to other aspects of their data. Certainly, it begins to shake my confidence in the project. Am I making an elementary unzipping mistake or are the database managers careless? Because the credibility of the data is at risk, it is important to address even minor concerns. I recommend easy remedies for the authors. They should enlist a set of volunteers with different computer expertise and hardware to test the accessibility of their data files. It should also be a simple matter to find a native English-speaking paleontologist who can quickly correct the English grammar and syntax, given an editable text file. I can correct English grammar, but I am worried about database managers and editors whose revised datafiles are unreadable on a very standard Windows computer that can still read the previous version.

Author reply:

Duplicate part of data description, of the whole manuscript, was deleted. Contents were re-organized. And the English wording was checked by software (Grammar) and authors. Graptolite experts who help curating specimens are mentioned in acknowledgments part. Opinion data, or comment from different authors are emphasized and was merged into the dataset.

---

## Author Response (AR3)

Title: A multi-dimensional dataset of Ordovician to Silurian graptolite specimens for virtual examination, global correlation and shale gas exploration
Author(s): Hong-He Xu et al.

Authors' response to reviewing, underlined text, by Honghe Xu (hhxu@nigpas.ac.cn)

Reviewer's Background and Summary Review:
I was trained as a taxonomist, but for trilobites and conodonts, not graptolites. I read graptolite systematics extensively as I build and maintain a high-resolution, age-calibrated, global time-line of the species-level macroevolution of the entire graptolite clade. My composite timeline has supported age calibration of the Ordovician and Silurian time scales in addition to several macroevolutionary studies. For the time-line project, I wrote my own data-management software that supports the stratigraphic sequencing program that I co-wrote. Both programs are in Fortran.
Although I rely almost entirely upon co-authors to validate our selection of the graptolite publications that I compile and to vet the quality of the taxonomy, it is likely that my experience is above-average among the potential users that the authors describe for their database. The database should surely be enormously valuable for my time-line projects. I trust this statement clarifies which components of this manuscript that I might be best qualified to review.
Reply: the reviewer, of course is qualified, and kindly give many constructive suggestions and helped greatly improve the manuscript.

This is the fourth version of the image database that I have tested. The authors have made a succession of substantive and successful changes. My previous struggles with the download speed and file format have been essentially alleviated in this latest version. The database is surely ready for a wide user population.
Reply: thank the editor and the reviewer.

The bulk of this third review focuses on the accompanying manuscript. I will deal with its content. to some extent, but focus primarily on the quality of the English text. The troublesome errors are not substantively improved by the latest edits. Although the quality of English written by non-native English speakers should not reflect upon the quality of the database, I fear that readers might lose confidence in the authors' attention to detail in the database too. That would be most unfortunate; the database surely deserves better.
In most respects, my prior criticism of the text is still valid. Previously I mentioned the common types of grammatical errors and poor word choices. The one example of subject-verb disagreement that I cited from the introduction was not fixed. The authors' reply mentions grammar-checking software; it seems to be quite inadequate. This time I have added a marked-up a copy of the PDF file with more edits, corrections, suggestions and questions to guide the authors' revisions.
Reply: in this version we accept reviewer's suggestion and annotated PDF file. The writing is obviously improved. We also have the manuscript checked by colleagues.

In summary, the database files are ready for use by experts, amateurs and students, just as the authors explicitly intend. The manuscript is not yet ready for publication. Obvious inattention to detail in the text file could undermine users' confidence in the authors' quality control in the database itself.

Reply: thank the reviewer, in the revision, we note this problem. Opinion or comment only represent a few authors' view point. Actually, we give only comments of 1-2 graptolite experts, of which might not be accepted by others. So, in the revision we claimed this point. We respect the tagged name (label) of the individual specimen. Our dataset only provides the platform or access to researchers or ones who are interested in our fossils. However, the revised comments of some specimens are recorded in our uploaded excel file. We can also make related explanation in the manuscript.

Importance of the Image Database

Paleontologic research has been advanced considerably by on-line availability of larger collections of publications than are housed as paper copies in most institutional libraries. This image database is a major step toward an exciting parallel development for paleontological museum collections. The on-line images will not entirely replace expensive travel to examine unique physical specimens or the risky loaning of unique specimens via mail services. They will surely, however, allow more effective preliminary evaluations than the limited photographic plates in printed journals.

Reply: thanks. We put this as one of the key contributions of this study. We will go on to enlarge and improve our database.

Revisions of the Image Database

The large image collection is not set up for on-line browsing. Instead, users download the files to their own computers. Many users will surely welcome this. Three substantive changes to the database structure have made the downloading and searching of the database faster and easier.

1. The image collection has been divided into 49 zipped folders. Individual folders can be downloaded in a fraction of the time that was needed for the single folder; such long download times tended to crash personal computers, even with robust institutional connections to the internet.

2. An intermediate problem with the naming of compressed folders has been corrected. Folder contents can now be extracted by basic components of common operating systems.

Reply: thank the editor's work

3. The folder names and file names now include genus names to indicate the folder and file contents. Users no longer need to browse through numerous folder contents to locate species of interest. This elegant convention obviates my suggestion that the xlxs spreadsheet could add a column that mapped specimens to the appropriate folder. I thank the editor, Kirsten Elger, for guiding me and the authors through these

successive improvements.
Reply: the spreadsheet file and image folders were both improved for the sake of user's convenience.

Most of the image files are paired. An image of the entire rock sample now includes a ruler for scale. For the close-up images of the graptolite specimens on the rock surface, the dimensions of morphologic features can readily be estimated by comparison. Earlier versions did not provide scales and this was not possible.
Reply: this was done and greatly improve the uploaded dataset. I believe this is one of the key contributions or innovations of our work.

The Accompanying Manuscript Document
I have annotated the PDF file with edits, questions and suggestions. Many simply involve subject-verb agreement or the use of definite and indefinite articles (the, an, a). Here are some more significant issues.
Reply: the annotated file is very useful to improve the text. Most suggestions were followed in the revised manuscript.

1. Some species names have been emended from the original publication. Either an erroneous published identification has been corrected, or the published name has been synonymized with a previously named species, or the specimen has been re-assigned to a newer species that was established after publication of the paper to which the museum specimen is connected. The spreadsheet provides both names, emended and "tagged," but does not distinguish between the three possible reasons to emend the name. Some of this uncertainty could be clarified if both the names included author and year. Those two terms are, after all, requirements of a valid taxon name. Readers may be unsure what the authors mean by "tagged" in this context.
Reply: this is quite important. We noted this point and make explanations in the text. Emendation or academic correction is relative and quite personal. We do not want to emphasize this in the dataset description study. Our focus should be the dataset itself.

2. The manuscript states that paleontologists (plural) have provided these taxonomic corrections or updates. Only one author (XM) is credited with this kind of contribution; nobody else is credited for this in the acknowledgements. It would seem that any user wishing to reference an emended name in a synonymy list would, therefore, need to credit the name to the current authors; i.e. Xu et al. 2022. No detailed justifications for the amendments are provided in the xlxs file.
Reply: same answer to the previous one.

3. Users of the database might wish to construct or correct range charts for the localities. They would need to know whether an emended name applies only to one specimen from that locality, to all specimens at one horizon, or to all uses of the name in the cited reference. This kind of uncertainty can arise from traditional publications too. It is difficult to update range charts, but frequently desirable.

Reply: the dataset with detailed information of every specimen is the first and quite crutial step. We then develop the visualizer model (software) to fulfil the functions of data querry and retrieval.

4. Some users might not have easy access to the cited Chinese literature. Is it possible for the database to include at least the systematic taxonomy sections from those papers? I imagine there would be copyright issues.
Reply: this will also be fulfilled in the visualizer model.

5. The manuscript has the problem of addressing experts, amateurs and students – three stated potential users of the database with potentially very different levels of sophistication. Some parts, like the first paragraph, are not needed for experts and tend to be oversimplified for students. Other parts omit advanced considerations, like the formal treatment of synonymy and revision.
Reply: now we are more clear the purpose and significance of our study. It is a data description study. It provide the data access and also multiple usage of the dataset. This part was revised according to reviewer's suggestions.

---

## Author Response (AR4)

Revision notes

Manuscript: essd-2021-280
Title: A multi-dimensional dataset of Ordovician to Silurian graptolite specimens for virtual examination, global correlation and shale gas exploration
Author(s): Hong-He Xu et al.
MS type: Data description paper
Correspondence: H.-H. Xu (E-mail: hhxu@nigpas.ac.cn)
Reviewer and/or editor reports:
Public justification (visible to the public if the article is accepted and published):
Dear Honghe,
Happy new year!!
I am pleased to have received the new review report of Peter Sadler. The main point is that still needs to be addressed, before I can finally accept the paper, is to make the dissertation of Ma (2020) available for the general public. This step is crucial, because you refer to this (unpublished) thesis various times in the manuscript as reference for taxonomical changes. The reader must be able to access this thesis to understand the reason for the change of the known taxonomy. Do you see any chance to make the thesis available? Technically, you will need to get the approval of the university at which the thesis was done. Then I would suggest to publish the thesis either at the respective university or at an alternative text repository. If you need guidance for this, please don't hesitate to contact me.
best regards,
Kirsten Elger

Revision Reply (by H.H Xu):
The manuscript ESSD-2021-280 has been prepared since the mid of 2019 and the first version was finished at the end of 2020. During our preparing Ma Xuan was also in my Institute and doing her dissertation, that is why she is in our author list. The reference we cited as '(Ma, 2020), actually, has been published formally. It is in Chinese with English summary (long summary, meeting the normal academic paper requirements). The part of mentioning 'unpublished' was changed accordingly. I understand that it might be difficult to search the dissertation to some authors who are outside China, for the index is in Chinese. One can't find it through Google scholar either. I believe that it will easily findable soon, after all, the dissertation has been published for 3 years. Alternately, a note after the reference list is added as '[for formal publication of academic using, please contact the present author H.-H. Xu]' (lines 260-261).
I noted that synonym information of our dataset seems readily causing misunderstanding. Naming and synonyms are really beyond the present dataset study. we here emphasize the using tagged name of specimens for searching, and that revisions represent experts comments and might need formal study (lines 165-169).
Some spelling mistakes are corrected, in lines 53, 81, 121, 125, 193, 309, 355.
Some parts are re-worded, 'who is interested in fossils' to 'who is interested in

palaeontology' (line 142)

Reference 'Xu et al., in press', changed to '2022' (line 152). And its full citation is given in line 277.

In reference part, add 'and' before the last author's name. Use 'et al.' after the third name, instead of listing all authors (see lines 244-292).

---

## Author Response (AR5)

Revision notes

ESSD-2021-280
Received: 16 Aug 2021 – Discussion started: 12 Nov 2021 – Revised: 28 Jan 2023
A multi-dimensional dataset of Ordovician to Silurian graptolite specimens for virtual examination, global correlation and shale gas exploration
Authors: Hong-He Xu et al.
Corresponding E-mail: hhxu@nigpas.ac.cn

Dear Honghe,

I am very pleased to hear that you have received permission to publish the PhD thesis of Ma that is highly relevant for this manuscript. You may add it to the Zenodo repository that presents the data or as individual publication via Zenodo that is linked to the data DOI (via relatedIdentfier "is Documented By". Please let me know if you want any assistance from me for the data update. We could meet via Zoom and share screens if you like.

[reply: a DOI is assigned to this thesis. It is cited as following and revised in the manuscript

Ma, X.: Palaeontology, biostratigraphy and palaeoecology of the graptolite from the Hulo Formation (Darriwilian – Sandbian) in northwestern Zhejiang Province, East China. A Ph.D dissertation submitted to University of Chinese Academy of Sciences (supervised by Prof. Zhang Y-D). 1-301. 2020. DOI:10.5281/zenodo.7827023.

]

The current version of the manuscript still includes the old links to the data. May I kindly ask you to ONLY use the following DOI and citation for the data throughout the manuscript (in the abstract, data availablity and references sections)

https://doi.org/10.5281/zenodo.6688670

Xu, H.: An image dataset of 1550 Ordovician to Silurian graptolite specimens for correlation and shale gas exploration, https://doi.org/10.5281/ZENODO.6688670, 23 June 2022.

[DONE!]

If you decide to publish Ma's thesis not with the data, you will need to add another citation to the references. If you add the thesis to the existing Zenodo repository, you will automatically get a new DOI for the new version. The DOi and citation above, however, will remain valid as it represents always the latest version.

many thanks and best regards,
Kirsten Elger

Additionally, according to the editorial suggestion (Email of editorial@copernicus.org, 30-Jan-2023), original figure 6 is changed to Table 1. Figures and table are re-labelled in the revised manuscript accordingly.